# Decision of Anticoagulation in Nonvalvular Atrial Fibrillation in the Real World in the Non-Antivitamin K Anticoagulants Era

**DOI:** 10.3390/healthcare10071333

**Published:** 2022-07-18

**Authors:** Gabriela Silvia Gheorghe, Andreea Simona Hodorogea, Andrei Cristian Dan Gheorghe, Dragoș Emanuel Popa, Simona Vulpe, Cristina Georgescu, Ruxandra Bănică, Andrei Gorgian Florescu, Elena Cristiana Trușcă, Omer Eden, Ana Ciobanu, Irina Pârvu

**Affiliations:** 1Cardiology and Internal Medicine Department, Theodor Burghele Clinical Hospital, 050653 Bucharest, Romania; gsgheorghe@gmail.com (G.S.G.); ana.ciobanu@umfcd.ro (A.C.); chivuirina@yahoo.com (I.P.); 2Faculty of Medicine, Carol Davila University of Medicine and Pharmacy, 020021 Bucharest, Romania; acd2403@yahoo.com (A.C.D.G.); dragos-emanuel.popa@rez.umfcd.ro (D.E.P.); simona-maria.vulpe@rez.umfcd.ro (S.V.); crist70el@gmail.com (C.G.); banica.ruxandra95@yahoo.com (R.B.); andxro@gmail.com (A.G.F.); elena-cristiana.trusca@rez.umfcd.ro (E.C.T.); omer.eden.94@gmail.com (O.E.)

**Keywords:** atrial fibrillation, antivitamin K anticoagulants, non-antivitamin K anticoagulants, CHA2DS2-VASc, HASBLED, CCI

## Abstract

*Background.* Patients with nonvalvular atrial fibrillation (NVAF) have five times higher risk of stroke than the general population. Anticoagulation (ACO) in NVAF is a class I indication after assessing the CHA2DS2-VASc and HAS-BLED scores. However, in the real world, NVAF patients receive less ACO than needed due to patients’ comorbidities that can be assessed by the Charlson comorbidity index (CCI). The use of non-antivitamin K anticoagulants (NOAC) has improved the decision to anticoagulate. *Objective*. We analyzed the factors influencing the ACO prescribing decision in NVAF patients in the real world and the changes induced by the introduction of NOAC. *Method*. We carried out an observational retrospective cross-sectional study that included consecutive patients with permanent NVAF and CHA2DS2-VASc ≥ 2, admitted to a community hospital between 2010–2011 (group 1, 286 patients), when only vitamin K antagonists (VKA) were used, and 2018–2019 (group 2, 433 patients), respectively. We calculated CHA2DS2-VASc, HAS-BLED, and CCI and recorded the ACO decision and the use of VKA or NOAC in group 2. We compared the calculated scores between ACO and non-anticoagulated (nonACO) patients in both groups and between groups. *Results*. A 31.5% share of patients in group 1 and 12.9% in group 2 did not receive ACO despite a CHA2DS2-VASc score ≥ 2. In group 1, nonACO patients had higher HAS-BLED and CCI scores than the ACO patients, but their CHA2DS2-VASc scores were not significantly different. Old age, dementia, severe chronic kidney disease, neoplasia, and anemia were the most frequent reasons not to prescribe anticoagulants. In group 2, more nonACO patients had dementia, diabetes mellitus, and higher HAS-BLED than ACO patients. Moderate-severe CKD, neoplasia with metastasis, liver disease, anemia, and diabetes mellitus were statistically significantly more frequent in nonACO patients from group 1 than those from group 2. In group 2, 55.7% of ACO patients received NOAC. *Conclusions*. In real-world clinical practice, the decision for anticoagulation in NVAF is influenced by patient age, comorbidities, and risk of bleeding, and many patients do not receive anticoagulants despite a high CHA2DS2-VASc score. The use of NOAC in the past few years has improved treatment decisions. At the same time, the correct diagnosis, treatment, and surveillance of comorbidities have cut down the risk of bleeding and allowed anticoagulant use according to guidelines.

## 1. Introduction

### 1.1. Background

Nonvalvular atrial fibrillation (NVAF) is the most common sustained cardiac arrhythmia in adults, with a prevalence of 9% in those aged 80 years or older [1,2] and a five times higher risk of developing stroke than in the general population [2]. According to guidelines, anticoagulation in NVAF is a class I indication of treatment after assessing CHA2DS2-VASc as a stroke score and HAS-BLED as a hemorrhage score [1]. The guidelines also advise the regular reassessment of both scores to identify changes in the modifiable stroke and hemorrhagic risk factors and the reallocation of patients. Unfortunately, common medical factors are involved in stroke and hemorrhagic risk, especially in the elderly. According to the AFFIRM study [3], increased age, heart failure, hepatic or renal disease, diabetes, first NVAF episode, warfarin use, and aspirin use were significantly associated with major bleeding events.

Hylek EM et al. [4] analyzed the hemorrhagic risk under warfarin in 472 patients with atrial fibrillation over 65 years old and reported a cumulative incidence of major hemorrhage of 13.1 per 100 person-years in patients over 80 years old and 4.7 for those under 80 years. On the other hand, the BAFTA study [5] demonstrated that the use of aspirin instead of warfarin in patients with atrial fibrillation aged 75 years or over was not superior in preventing ischemic stroke and systemic embolism or intracranial or extracranial hemorrhage. Therefore, according to guidelines, NVAF patients with high bleeding risk must receive anticoagulation under more frequent medical monitoring [1].

Despite these recommendations, in real-world medical practice, a significant number of NVAF patients are non-optimally anticoagulated due to patient-dependent factors and physician decisions. Calderon JM et al. [6] demonstrated in 123,227 NVAF patients that 11.7% of those with CHA2 DS2-VASc ≥ 2 did not receive anticoagulant therapy. Some patients could have a higher hemorrhagic risk than predicted, related to the presence of comorbidities not included in the HAS-BLED score but accounted for in the Charlson comorbidity index (CCI). On the other hand, data have changed in the past years due to the availability of the non-antivitamin K anticoagulants (NOAC), which are not inferior regarding efficiency and are safer in terms of bleeding risk than vitamin K anticoagulants (VKA) in NVAF [7]. Waranugraha Y et al. [8] performed a meta-analysis that included 2,287,288 NVAF patients and found that NOACs reduced the stroke risk and intracranial bleeding risk more than warfarin, but not the gastrointestinal bleeding risk. NOACs also lowered all-cause mortality risk. Therefore, their use in clinical practice has increased in recent years.

### 1.2. Objective

We aimed to analyze the factors that influence in real-world clinical practice the decision to prescribe anticoagulant treatment in patients with NVAF and the changes in practice induced by the introduction of NOAC as a therapeutic option.

## 2. Materials and Methods

We performed an observational retrospective cross-sectional study that included consecutive patients with permanent NVAF and CHA2DS2-VASc ≥ 2 admitted to a community hospital between 2010–2011 (group 1) and between 2018–2019 (group 2). The analysis was focused on the anticoagulation decision upon discharge, made according to the individual clinical data, as this decision determined the therapeutic option in ambulatory practice. The attending cardiologists established the diagnosis of NVAF in each case according to current guidelines [1]. We evaluated the medical records of each patient, recorded the clinical data that could influence the decision to anticoagulate, and calculated the CHA2 DS2-VASc score [9], HAS-BLED score [10,11], and CCI (Table 1) [12,13]. We recorded the decision to implement anticoagulants upon discharge and correlated it with the aforementioned scores and, in group 2, with the use of VKA or NOAC.

Data are presented as mean ± standard deviation for numerical variables and as absolute numbers and percentages for categorical variables. For numerical variables, parametric (Student’s *t*-test) or non-parametric (Mann–Whitney, Kruskal–Wallis) tests were used, according to the distribution of data. Additionally, Levene’s test was used for assessment of the homogeneity of variances. Chi square test and Fisher’s exact test were used to compare categorical variables. Multivariate logistic regression was used for analysis of the association of the different medical factors with the prescription of anticoagulants at discharge.

## 3. Results

There were 286 patients analyzed in group 1, and 433 patients in group 2 (Figure 1).

The demographic data are presented in Table 2. The mean age distribution was similar in both groups (*p* = 0.18), and there were more men in group 1 (*p* = 0.047). Compared to group 2, in group 1 dementia, (*p* < 0.0001), COPD (*p* = 0.0191), moderate-severe kidney disease (*p* < 0.0001), mild liver disease (*p* < 0.0001) and neoplasia (*p* = 0.036) were more prevalent, and peripheral artery disease (PAD) (*p* = 0.049), cerebrovascular disease (ASC), (*p* = 0.0002), diabetes mellitus (DM) with end-organ damage (*p* = 0.005), and arterial hypertension (HTN) (*p* = 0.0002) less prevalent. History of cerebrovascular disease was represented in both groups mainly by ischemic stroke, with a single case of hemorrhagic stroke in group 2. HAS-BLED score and CCI were significantly higher, and CHA2 DS2-VASc score was significantly lower in group 1 than in group 2. Upon discharge, 199 (68.5%) patients in group 1 and 377 (87.1%) patients in group 2 received the indication of ACO (*p* < 0.0001), and 87 (31.5%) patients in group 1 and 56 (12.9%) in group 2 were nonACO (*p* < 0.0001) (Figure 1). In group 2, 210 (55.7%) ACO patients received NOAC. Those who were prescribed VKA were younger than those treated with NOAC (*p* = 0.03), with a tendency, without reaching statistical significance, to have higher HAS-BLED and CCI scores compared to NOAC patients (Table 3).

In group 1, nonACO patients were older than ACO patients (*p* = 0.001), without difference in male: female distribution. NonACO patients had significantly more frequent ASC (*p* = 0.036), moderate-severe CKD (*p* = 0.034), neoplasia with and without metastasis (*p* = 0.004, respectively *p* = 0.021) and anemia (*p* < 0.0001). Three nonACO patients and no ACO patients had leukemia. The CHA_2_ DS_2-_ VASc score was not statistically significantly different between ACO and nonACO patients (3.9 ± 1.7 in ACO, 4.2 ± 1.5 in nonACO, *p* = 0.138). HAS BLED score (2.7 ± 0.9 in nonACO and 2.4 ± 1.1 in ACO patients, *p* = 0.0135) and CCI (7.3 ± 3.2 in nonACO patients and 5.7 ± 2.5 in ACO patients, *p* = 0.0002) were significantly higher in nonACO patients than in ACO patients (Figure 2).

In group 2, there was no age or gender difference between nonACO and ACO patients. Compared to ACO patients, nonACO patients had a higher prevalence of dementia (*p* < 0.0001), uncomplicated DM (*p* = 0.01), and a lower prevalence of congestive heart failure (*p* = 0.044). CHA2 DS2 -VASc score was 4.5 ± 1.6 in ACO and 4.5 ± 1.9 in nonACO patients (*p* = 0.8934). HAS-BLED score was significantly higher in nonACO patients (2.1 ± 1.3 in ACO, and 2.6 ± 1.4 in nonACO, *p* = 0.0294). CCI was similar between ACO and nonACO patients (Figure 2).

Multiple logistic regression was used to analyze the relationship between the different medical factors and the prescription of anticoagulants at discharge. The models included variables associated with the outcome of interest in bivariate analysis at a *p*-value < 0.20 and age because of its clinical relevance. From the factors assessing renal function, we chose the estimated glomerular filtration rate as covariate in the models. The results are depicted in Table 4 and Table 5. After performing multiple logistic regression, in group 1, anemia was an independent predictive factor for the decision not to anticoagulate at discharge (OR = 3.20, 95%CI (1.60;6.40), *p* = 0.001). In group 2, the presence of dementia was an independent predictive factor for the decision not to anticoagulate at discharge (OR 4.15, 95% CI (1.37,12.55), *p* = 0.01), and congestive heart failure independently predicted the prescription of anticoagulants at discharge (OR = 2.5, 95%CI (1.22,5.12), *p* = 0.01). Age was an independent predictor for NOAC prescription (OR = 1.03, 95%CI (1.00,1.06), *p* = 0.01), while CCI predicted VKA prescription (OR = 1.22, 95%CI (1.02,1.46), *p* = 0.02).

## 4. Discussion

Atrial fibrillation is responsible for 20–30% of all ischemic and 10% of cryptogenic strokes, and VKA therapy reduces stroke risk by 64% and mortality by 26% [1]. However, there is an increased bleeding risk, and the favorable results depend on the persistence of INR values in the therapeutic range more than 70% of the time. The therapy must be monitored by frequent blood draws, and comorbidities can increase the hemorrhagic risk. The CHA2 DS2-VASc and HAS-BLED scores are useful for the decision of anticoagulation, and the recommendation is to treat the patients according to the stroke risk but to monitor them more frequently if the bleeding risk is high. However, there are comorbidities related to a high bleeding risk that are not included in the HAS-BLED score, and in real-world clinical practice, many patients with NVAF do not receive ACO. CCI includes medical comorbidities related to one- and ten-year mortality risk and can influence the anticoagulation decision in NVAF. Since the introduction of NOAC to prevent thrombosis in NVAF, there is no need for INR monitoring, and there are fewer drug and food interactions. Furthermore, under NOACs, there is a 51% reduction in hemorrhagic stroke risk, a similar ischemic stroke risk reduction compared with VKAs, and a 10% reduction in all-cause mortality [14].

We identified medical factors influencing the decision not to anticoagulate the patients with NVAF despite CHA2DS2 -VASc score ≥ 2 during two periods, one group before and the other after the introduction of NOAC in clinical practice. We observed that 31.5% of NVAF patients admitted to the hospital in an era when only VKAs were used as oral anticoagulants, were not anticoagulated at discharge despite CHA2DS2-VASc score ≥ 2. The nonACO patients had more elevated HAS-BLED scores and CCI than ACO patients. However, their CHA2DS2-VASc scores did not significantly differ, suggesting that in real-world practice, the anticoagulation decision is highly dependent on the hemorrhagic risk. Old age and the association of NVAF with dementia, severe chronic kidney disease, neoplasia, and anemia were the most frequent factors influencing this decision. In fact, the mortality rate associated with anticoagulation-related major bleeding is as high as 13.4% [15]. The multivariate analysis showed that in our study group, anemia was an independent factor associated with the decision not to anticoagulate. Scientific data demonstrate that anemia is frequently associated with the decision to refrain from anticoagulation in patients with NVAF and poor thromboembolic prophylaxis in anticoagulated patients. Bonde A.N. et al. [16] investigated the risk of bleeding and thromboembolic events in 18,734 Danish patients diagnosed with NVAF between 1997 and 2012. Of the patients in that study, 14% had moderate/severe anemia, and 22.4% received VKA, compared to 78.1% of patients with no or mild anemia. Patients with moderate/severe anemia had a 5.3% increased absolute risk of major bleeding, 9.1% lower median time in the therapeutic range on VKA, and no reduced risk of stroke or thromboembolic events compared with nonanemic patients. Stopping the anticoagulation because of a hemorrhagic event is associated with an increased risk of all-cause mortality, ischemic stroke, or systemic embolism, as shown in the Garfield-AF Registry [17]. Hess PL et al. [18] demonstrated that 23% of the NVAF patients in the ORBIT-AF study were not anticoagulated, and the independent factors associated with this decision were the type of atrial fibrillation, left atrial diameter enlargement, and age over 80. In that study, nonanticoagulated patients had a higher risk of death. According to guidelines [1], a high bleeding risk score should not lead to withholding ACO, but the bleeding-modifiable risk factors should be corrected and the patient should be reassessed more frequently.

Our data regarding the use of VKA in NVAF patients were concordant with data in the literature. Ogilvie et al. [19] performed a meta-analysis of 98 studies published between 1997–2008 regarding the use of VKA in patients with NVAF and demonstrated that patients with CHADS2 score ≥ 2 were sub-optimally treated, with seven of nine studies reporting treatment levels between 39–92.3%.

Patients in group 2 had higher CHA2DS2-VASc scores and lower HAS-BLED and CCI scores than in group 1. As comorbidities, they had a higher prevalence of PAD, ASC, DM with end-organ damage, and HTN, and a lower prevalence of dementia, COPD, moderate-severe kidney disease, mild liver disease, and neoplasia than patients in group 1. Of the patients in group 2, 87.1% received ACO, 44.29% of them received VKA, and 55.7% NOAC. Patients on NOAC were older than patients on VKA, and patients on VKA had higher CCI and HASBLED scores, but at the limit of statistical significance. This option could reflect the long-term habit of doctors to use VKA despite extensive proof of the safety and efficacy of NOAC [14,20]. However, the high proportion of patients treated with NOAC in group 2 demonstrated the changes in this habit. Of the patients in group 2, 12.9% were nonACO with higher HAS-BLED scores than ACO patients in the same group, which could explain this decision. Dementia was independently associated with the decision to not prescribe anticoagulants in group 2. This could be due to the fear of lack of adherence to correct ACO administration despite the demonstrated protection against the occurrence and aggravation of dementia in patients with NVAF under ACO [21]. Heart failure was an independent factor associated with the decision to anticoagulate. In fact, in group 2, the prescription of ACO was more adherent to the guidelines’ recommendations [1]. Many studies demonstrated the usefulness of NOAC in NVAF, with lower discontinuation rates than in patients with VKA [14,20]. A meta-analysis by Ozaky AF et al. [22] regarding the use of NOAC in NVAF described good adherence to the therapy in 66% of patients and persistence in 69% of patients, higher with NOAC than with VKA. These observations are important, because the lack of persistence is associated with an increased risk of stroke [22].

*The limitations* of the study are multiple and result mainly from the retrospective design. Prescriptions of anticoagulants at discharge varied according to patients’ risk profiles and physicians’ and patients’ preferences, and the exact reasons have seldom been recorded in the medical records. Thus, our observations can only establish association and not cause–effect relationships between the clinical factors and scores and the decision to anticoagulate. In addition, we had no follow-up data on ambulatory decision changes or outcomes after discharge. Being a single-center study, the study groups may not be representative of the population.

## 5. Conclusions

In the real world, the anticoagulation decision in NVAF is influenced by patients’ age, comorbidities, and risk of bleeding. Many patients do not receive anticoagulation despite a high CHA2DS2-VASc score. The use of NOAC in the past few years has improved the treatment decision. At the same time, the correct diagnosis, treatment, and surveillance of comorbidities has cut down the risk of bleeding and allowed anticoagulation according to guidelines.

## Figures and Tables

**Figure 1 healthcare-10-01333-f001:**
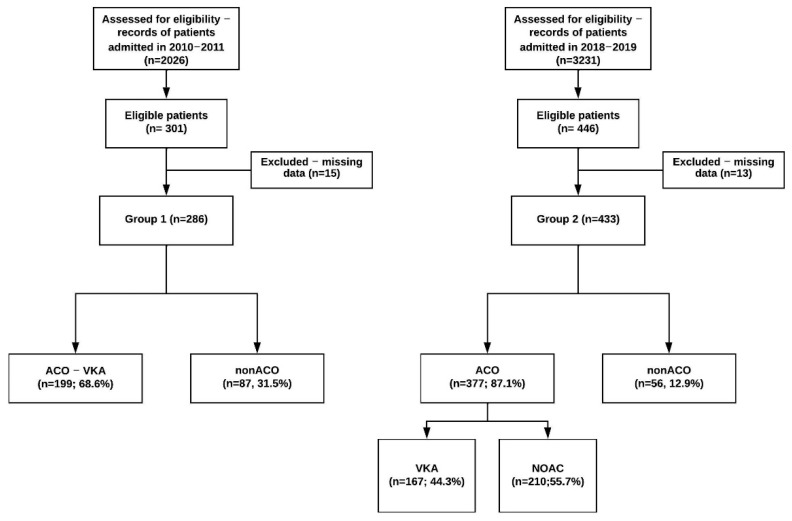
Flowchart of the patients included in the study. Pts = patients; ACO = with anticoagulation upon discharge; nonACO = without anticoagulation upon discharge, VKA = vitamin K antagonists.

**Figure 2 healthcare-10-01333-f002:**
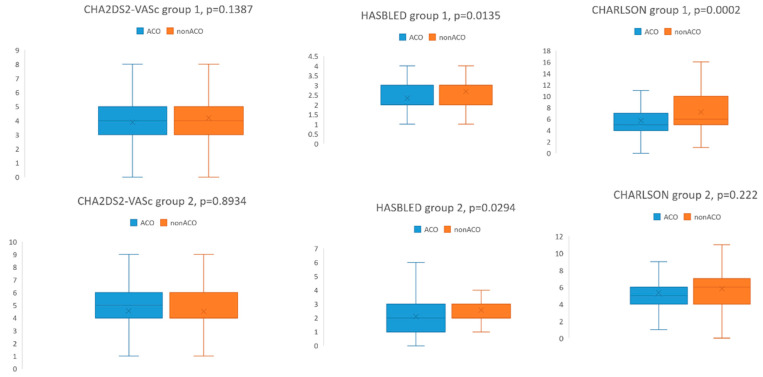
CHA_2_ DS_2-_ VASc, HASBLED and Charlson comorbidity index in anticoagulated (ACO) and nonanticoagulated (nonACO) patients from group 1 and group 2, respectively.

**Table 1 healthcare-10-01333-t001:** Charlson comorbidity index (CCI) and the risk of one-year and ten-year mortality [10].

Condition	Point	1-Year Rate of Mortality (Validated in 559 Patients)		10-Year Rate of Mortality (Validated in 685 Patients)	
Myocardial infarction	1	Scores	Rate (%)	Scores	Rate (%)
Chronic heart failure	1	0	12	0	8
Peripheral vascular disease	1	1–2	26	1	25
Cerebrovascular disease	1	3–4	52	2	48
Dementia	1	≥5	85	≥3	59
Chronic pulmonary disease	1				
Connective tissue disease	1				
Ulcer disease	1				
Mild liver disease	1				
Diabetes	1				
Hemiplegia	2				
Moderate or severe renal disease	2				
Diabetes with end organ damage	2				
Any tumor without metastasis	2				
Leukemia	2				
Lymphoma	2				
Moderate or severe liver disease	3				
Metastatic solid tumor	6				
Acquired immunodeficiency syndrome (AIDS)	6				

One point is added for patients aged 41–50 years; 2 points for those aged 51–60 years; 3 points for those aged 61–70 years; 4 points for those 71 years or older.

**Table 2 healthcare-10-01333-t002:** Demographic and clinical data of patients included in the study between 2010–2011 (group 1) and 2018–2019 (group 2).

Parameters	Group 1, 286 pts	Group 2, 433 pts	*p*
Age (years)	72.8 ± 9.8	73.8 ± 10.4	0.1883
Sex—male	149 (52.1)	193 (44.6)	**0.0479**
Myocardial infarction	41 (14.5)	57 (13.2)	0.6097
Congestive heart failure	213 (74.5)	317 (73.2)	0.7822
PAD	22 (7.7)	53 (12.2)	**0.0496**
ASC	66 (23.2)	156 (36)	**0.0002**
Dementia	43 (15)	23 (5.3)	**<0.0001**
COPD (*n*, %)	67 (23.4)	71 (16.4)	**0.0191**
Uncomplicated DM	52 (18.2)	78 (18)	0.9665
Complicated DM	22 (7.8)	63 (14.5)	**0.0053**
Moderate-severe CKD	81 (28.6)	48 (11.1)	**<0.0001**
Hemiplegia	6 (2.1)	7 (1.6)	0.6309
Leukemia	3 (1)	4 (0.9)	1
Lymphoma	2 (0.7)	2 (0.5)	0.6498
Neoplasm	47 (16.5)	48 (11.1)	**0.0364**
Metastasis	11 (3.9)	15 (3.5)	0.7813
Mild liver disease (*n*, %)	102 (35.8)	30 (6.9)	**<0.0001**
Severe liver disease (*n*, %)	11 (3.9)	13 (3)	0.5393
AIDS (*n*, %)	0 (0)	0 (0)	1
HTN (*n*, %)	216 (77.9)	381 (88)	**0.0002**
Anemia (*n*, %)	69 (30.1)	140 (32.3)	0.5621
Creatinine (mg/dL)	1.2 ± 0.5	1.2 ± 0.5	0.6362
ClCr by MDRD equation (mL/min/1.73 m^2^)	60.7 ± 20.3	61 ± 20.9	0.8743
ACO at discharge (*n*, %)	199 (69.6)	377 (87.1)	**<0.0001**
Use of antiplatelet drugs (*n*, %)	72 (25.1)	115 (26.5)	0.67
CHA_2_ DS_2_-VASc score	3.9 ± 1.7	4.5 ± 1.6	**<0.0001**
HAS BLED score	2.5 ± 1.1	2.2 ± 1.3	**0.0001**
CCI	6.2 ± 2.8	5.4 ± 2.1	**0.0024**

ACO = oral anticoagulation; AIDS = acquired immunodeficiency syndrome; ASC = cerebrovascular disease; CCI = Charlson comorbidity index; CKD = chronic kidney disease; ClCr = creatinine clearance; COPD = chronic obstructive pulmonary disease; DM = diabetes mellitus; HTN = arterial hypertension; MDRD equation = Modification of Diet in Renal Disease equations; MI = myocardial infarction; PAD = peripheral arterial disease. ***p*****-values** marked with bold indicate statistically significant *p*-values.

**Table 3 healthcare-10-01333-t003:** Comparison between demographic and clinical data in nonvalvular atrial fibrillation patients on oral anticoagulation and non-anticoagulated in both groups and respectively between those anticoagulated with AVK or NOAC in group 2.

Parameter	Group 1 (*n* = 286)		*p*	Group 2 (*n* = 433)		*p*	Group 2 ACO (*n* = 377)		
	ACO (*n* = 199)	nonACO (*n* = 87)		ACO (*n* = 377)	nonACO (*n* = 56)		AVK (*n* = 167)	NOAC (*n* = 210)	*p*
Age (years)	71.6 ± 9.9	75.6 ± 8.7	**0.001**	73.7 ± 10.2	74.5 ± 11.4	0.5768	72.5 ± 10	74.7 ± 10.3	**0.03**
Sex—male (*n*, %)	106 (53.3)	44 (49.4)	0.5496	171 (45.4)	22 (39.3)	0.3936	74 (44.3)	97 (46.2)	0.71
Myocardial infarction (*n*, %)	30 (15.4)	11 (12.6)	0.5464	48 (12.8)	9 (16.1)	0.4953	16 (9.6)	32 (15.2)	0.10
Congestive heart failure (*n*, %)	147 (73.9)	66 (75.9)	0.7221	282 (75.2)	35 (62.5)	**0.0444**	132 (79)	150 (71.4)	0.08
PAD (*n*, %)	16 (8.04)	6 (6.9)	0.7384	42 (11.2)	11 (19.6)	0.0713	24 (14.3)	18 (8.5)	0.07
ASC (*n*, %)	39 (19.7)	27 (31)	**0.0366**	135 (35.8)	21 (37.5)	0.8057	61 (36.5)	74 (35.2)	0.79
Dementia (*n*, %)	26 (13.1)	17 (19.5)	0.1586	13 (3.5)	10 (17.9)	**<0.0001**	3 (1.7)	10 (4.7)	0.11
COPD (*n*, %)	48 (24.1)	19 (21.8)	0.6751	64 (16.7)	7 (12.5)	0.3985	32 (19.1)	32 (15.2)	0.31
Uncomplicated DM (*n*, %)	40 (20.1)	12 (13.8)	0.2032	61 (16.2)	17 (30.4)	**0.0103**	28 (16.7)	33 (15.7)	0.79
Complicated DM (*n*, %)	16 (8)	6 (6.9)	0.7574	53 (14.1)	10 (17.9)	0.4568	30 (17.9)	23 (10.9)	0.05
Moderate-severe CKD (*n*, %)	49 (24.9)	32 (37.2)	**0.0347**	41 (10.9)	7 (12.5)	0.7229	22 (13.1)	19 (9)	0.19
Hemiplegia (*n*, %)	4 (2)	2 (2.3)	1	5 (1.3)	2 (3.6)	0.2256	3 (1.7)	2 (0.9)	0.47
Leukemia (*n*, %)	0 (0)	3 (3.5)	**0.0277**	3 (0.8)	1 (1.8)	0.4265	1 (0.5)	2 (0.9)	0.58
Lymphoma (*n*, %)	0 (0)	2 (2.3)	0.0915	2 (0.5)	0 (0)	1	1 (0.5)	1 (0.4)	0.69
Neoplasm (*n*, %)	26 (13.1)	21 (24.1)	**0.0211**	39 (10.3)	9 (16.1)	0.2027	15 (8.9)	24 (11.4)	0.43
Metastasis (*n*, %)	3 (1.5)	8 (9.2)	**0.0041**	15 (3.9)	0 (0)	0.2357	6 (3.5)	9 (4.2)	0.73
Mild liver disease (*n*, %)	70 (35.4)	32 (36.8)	0.8168	24 (6.4)	6 (10.7)	0.2565	12 (7.1)	12 (5.7)	0.55
Severe liver disease (*n*, %)	5 (2.5)	6 (6.9)	0.0968	11 (2.9)	2 (3.6)	0.7945	6 (3.5)	5 (2.3)	0.48
AIDS (*n*, %)	0 (0)	0 (0)	1	0 (0)	0 (0)	1	0 (0)	0 (0)	1
HTN (*n*, %)	151 (79.1)	65 (75.6)	0.5182	334 (88.2)	47 (85.4)	0.5002	146 (87.4)	188 (89.5)	0.52
Anemia (*n*, %)	35 (21.9)	34 (49.3)	**<0.0001**	123 (32.6)	17 (30.4)	0.7348	57 (34.1)	66 (31.4)	0.57
Creatinine (mg/dL)	1 ± 0.5	1.2 ± 0.6	**0.0197**	1.1 ± 0.4	1.4 ± 0.8	0.1737	1.16 ± 0.53	1.08 ± 0.3	0.44
ClCr by MDRD equation (mL/min/1.73 m^2^)	63 ± 20.1	55.5 ± 19.8	**0.0086**	61.7 ± 20.1	55.8 ± 24.8	0.0982	60.6± 20.9	62.5 ± 19.4	0.36
CHA_2_ DS_2_-VASc score	3.9 ± 1.7	4.2 ± 1.5	0.1387	4.5 ± 1.6	4.5 ± 1.9	0.8934	4.5 ± 1.7	4.5 ± 1.5	0.97
HAS BLED score	2.4 ± 1.1	2.7 ± 0.9	**0.0135**	2.1 ± 1.3	2.6 ± 1.4	**0.0294**	2.24 ± 1.36	1.96 ± 1.22	0.06
CCI	5.7 ± 2.5	7.3 ± 3.2	**0.0002**	5.3 ± 1.9	5.8 ± 2.7	0.222	5.55 ± 1.92	5.17 ± 1.97	0.09

ACO = oral anticoagulation; AIDS = acquired immunodeficiency syndrome; ASC = cerebrovascular disease; CCI = Charlson comorbidity index; CKD = chronic kidney disease; ClCr = creatinine clearance; COPD = chronic obstructive pulmonary disease; DM = diabetes mellitus; HTN = arterial hypertension; MDRD equation = Modification of Diet in Renal Disease equations; MI = myocardial infarction; PAD = peripheral arterial disease. ***p*****-values** marked with bold indicate statistically significant *p*-values.

**Table 4 healthcare-10-01333-t004:** Logistic regression model of predictors of decision not to prescribe anticoagulants.

Covariate	Group 1			Group 2		
	Odds Ratio	95% Confidence Interval	*p*	Odds Ratio	95% Confidence Interval	*p*
Age	1.01	[0.96;1.05]	0.58	0.99	[0.95;1.02]	0.61
Anemia	3.20	[1.60;6.40]	0.001	-	-	-
Dementia	0.92	[0.36;2.34]	0.87	4.15	[1.37;12.5]	0.01
Neoplasia	1.38	[0.47;3.98]	0.55	-	-	-
eGFR	0.98	[0.96;1.00]	0.12	0.98	[0.96;1.00]	0.14
HASBLED	0.86	[0.58;1.27]	0.46	1.14	[0.89;1.46]	0.28
CCI	1.01	[0.83;1.23]	0.85	-	-	-
Severe liver disease	1.33	[0.24;0.85]	0.73	-	-	-
Congestive heart failure	-	-	-	0.39	[0.19;0.81]	0.01
PAD	-	-	-	1.32	[0.55;3.19]	0.52
DM	-	-	-	1.83	[0.85;3.9]	0.11

CCI = Charlson comorbidity index; DM = diabetes mellitus; MDRD equation = Modification of Diet in Renal Disease equations; PAD = peripheral arterial disease.

**Table 5 healthcare-10-01333-t005:** Logistic regression model of predictors of decision to use NOAC versus AVK in group 2.

Covariate	Odds Ratio	95% Confidence Interval	*p*
Age	1.03	[1.00;1.06]	0.01
Anemia	1.16	[0.71;1.88]	0.54
Dementia	1.22	[0.43;3.41]	0.69
Neoplasia	0.89	[0.41;1.90]	0.77
HASBLED	0.82	[0.68;0.99]	0.04
CCI	0.83	[0.70;0.98]	0.03
Severe liver disease	1.36	[0.28;6.45]	0.69
DM	1.04	[0.58;1.83]	0.88
Congestive heart failure	1.06	[0.62;1.79]	0.82

CCI = Charlson comorbidity index; DM = diabetes mellitus.

## Data Availability

The data underlying this article will be shared on request to the corresponding author.

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
