# Peer review of "Decision of Anticoagulation in Nonvalvular Atrial Fibrillation in the Real World in the Non-Antivitamin K Anticoagulants Era"

_healthcare, 2022, doi:10.3390/healthcare10071333_

Round 1
Reviewer 1 Report
The main purpose of the work is to compare oral anticoagulation prescription at discharge in pateints with non valvular AFIB in the VKA era and NOACS era. The main result is that prescription rate was significantly higher in the NOACS era and it was not influenced by renal failure , malignancy, cerebrovascular disease as in the VKA era.
There is some crucial data missing , for example concomitant antiplatelet therapy, the type of cerebrovascular disease ( ischemic vs. hemmorahgic).
In addition, performance of multivariate regression logistic analysis would deliniate independent predictors of non anticoagulation decision in both groups.
Reviewer 2 Report
1. This is retrospective analysis with relatively low number of patients.
2. Only one therapeutic decision point was taken into consideration. There is no data on follow up concerning anticoagulation decisions changes.
3. Based on available data is it possible to define the cause of decision not to implement anticoagulation in analyzed patients? Eg information in medical records why treatment was not introduced in particular patient. Presented data shows patients characteristics but not detailed decisions.
4. I do not agree with the sentence: “We analyzed the causes of the decision not to anticoagulate the patients with NVAF 183 despite CHA2DS2 -VASc score ≥ 2 and the changes of this decision after the introduction of 184 NOAC in clinical practice”. There are no data on “changes of this decision”. Author analyzed two different groups of patients but not the therapeutic decisions change in time as mentioned sentence suggest.
5. It will be interesting to see differences between VKA and NOAC in Group 2.
6. The limitations paragraph should be updated.
7. Conclusions: I am not sure that presented data describe factors influencing particular clinical decisions. Rather characteristics of patients receiving or not anticoagulants despite indications. We may conclude in which patients anticoag was not given but not which factor influence each one decision.
Round 2
Reviewer 2 Report
I have no more comments